# Positive-First Most Ambiguous: A Simple yet Efficient Active Learning Criterion for Novel Class Retrieval

## Abstract

Novel Class Retrieval is nowadays of crucial importance to leverage and explore the large amounts of available unlabeled data. It is defined as the iterative creation of a novel unknown class-of-interest based on an initial query, while relying on the use of human interaction. We formulate this problem as an Active Learning-based Relevance Feedback problem, where the human-in-the-loop periodically intervenes to label a subset of the data to train a one-versus-all classifier. In this case, the goal of the used Active Learning strategy is two-fold: rapidly fill the class-of-interest, and ensure that all class patterns are covered. However, most Active Learning methods only aim to improve the classifier performances, without considering the two previous aspects. To this end, we introduce a novel Active Learning criterion that balances classifier performances and class retrieval efficiency by selecting the most informative samples with the highest probability of being positive. We also formulate a novel coverage metric to evaluate retrieval performance. In addition to well-balanced datasets, evaluation is performed on real-world-like long-tailed datasets, which provide different degrees of class-of-interest imbalance. The results show that our criterion outperforms widely used strategies like *Most Ambiguous* and *Most Positive*. We also provide a framework to help researchers create and experiment with new Active Learning methods in the context of Novel Class Retrieval.

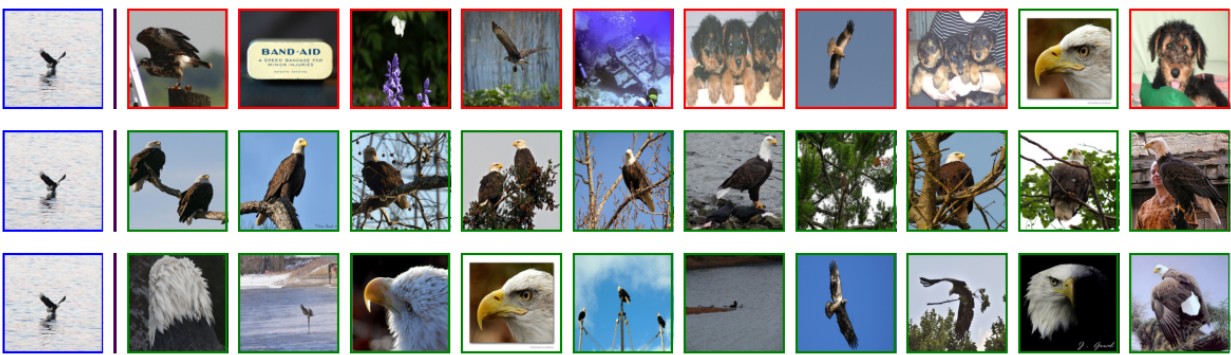

Figure 1: Retrieval results at iteration 5. The initial query image is in blue, the positive images in green and the negative ones in red. *Most Ambiguous* (top): too many negative images. *Most Positive* (middle): all positive images but very similar centered instances with very similar background. *Positive-First Most Ambiguous* (bottom): all positive diverse images with different backgrounds and dispositions.

## 1 Introduction

Over the last decades, there has been a remarkable increase in the volume and complexity of available data. Image data, in particular, are nowadays abundantly stored. However, the annotation process of these

vast amounts of data is both time-consuming and labor-intensive, which presents a crucial bottleneck when exploring large-scale datasets. Effectively leveraging this data calls for innovative techniques to identify and organize meaningful information.

One way users can effectively extract knowledge from such large unlabeled datasets is by searching for specific patterns and retrieving as many relevant samples as possible. In other words, the user of a large unlabeled database may be interested in creating a class that follows a certain pattern, say "dogs with hats", for which the use of off-the-shelf classifiers is inadequate, since predefined classifiers are not trained on such novel concepts. This problem setup is referred to as Novel Class Retrieval (NCR) Leroy et al. (2022). NCR addresses the problem of creating a new class, called class-of-interest, given an initial query, from a large unlabeled dataset. It assumes that this class has not been seen before and requires training a classifier to recognize it within the large data pool. NCR has two key characteristics: 1) it is *iterative*, as it continuously refines the classifier's knowledge about the desired concept, and 2) it is *interactive*, as it relies on human input to form this knowledge. As the retrieval progresses, the model presents some selected samples it deems *relevant* to the class-of-interest, and seeks further information about this class by querying the human oracle on their actual relevance. The user provides this Relevance Feedback through annotations. This setting is particularly well-suited to databases that are specific to an expert field and/or poorly represented on the web. Examples include audio-visual archives, digitized historical collections and naturalist databases.

Annotating thousands of examples from an unlabeled dataset is not only expensive, but also rather tiresome and laborious for the user. To mitigate this, the user's feedback should be limited to a very small subset of the data. Active Learning (AL) Settles (2009) is an excellent solution that allows to only select a subset of data, thus reducing the annotation cost. AL chooses the most *useful* samples to acquire labels for, according to an Active Learning criterion. This process is generally conducted in an iterative fashion, where batches of samples are labeled and incorporated into the training data of the classifier. Since our setup involves a human oracle, it is further constrained by a small annotation budget and limited wait times.

Most AL works present novel criteria that are used to select training data in a binary or multi-class classification context Settles (2009). In such a context, only the classifier performances are of interest to evaluate the AL criterion. In the NCR setting, the problem is rather a binary classification one, with the class-of-interest as the positive class, and the rest of samples as the negative one. However, classifier performance is not the only factor to consider. We want our criterion not only to 1) select useful samples to train a strong classifier, but also 2) prioritize samples that belong to the class-of-interest to improve the user's satisfaction, and 3) ensure such samples showcase the diversity of visual patterns of the class-of-interest by belonging to different "sub-concepts" within this class.

AL-wise, one category of AL criteria is based on uncertainty, or ambiguity, and performs well on the first objective but very poorly on the second. In addition, traditional AL methods have mainly been developed for relatively large annotation budgets which inherently allows for strong classifiers. Retrieval-wise, the strategy consisting on querying the most similar samples, or positive samples, checks the second objective but not so much the first one. Fig. 1 showcases the retrieval results of these criteria. Our main contribution is a simple but efficient AL criterion that checks all the evaluation boxes for our Novel Class Retrieval task. We call our criterion **Positive-First Most Ambiguous**, or **PF-MA**. PF-MA aims at combining the two previous selection strategies, by prioritizing the positive samples among the most ambiguous ones. This provides samples with sufficient information to train the classifier, but also with higher likelihoods of belonging the class-of-interest.

To check for the quality of the selected samples, classically used metrics Hameed et al. (2021); Srivastava et al. (2023) to evaluate the retrieval task do not take into account the spatial layout of the class-of-interest and the diversity of the samples. They are mostly based on the number of retrieved items, disregarding whether these items provide diversified information about the class. On the other hand, diversity metrics are mostly based on pairwise distances between the samples of the selected sets, or on the distribution of their similarity scores Friedman & Dieng (2022). Such metrics 1) only compute the diversity

of the selected set with regard to the whole feature space, not to the retrieved class, and more importantly 2) do not take into account the difference in the topological structures of the different classes. To this end, we formulate a novel metric, **class coverage**, to compute how much of the class-of-interest is actually covered through the retrieval process.

Another issue is that the class-of-interest might be of any size. This information is not available during the retrieval loop. Traditional image datasets are often near-perfectly balanced. This leads to a biased evaluation of the AL criterion as it is only tested on a certain class size. In this paper, we also aim at analyzing how different AL selection strategies perform on different class sizes and show how well our novel strategy performs in these conditions.

Our contributions can be summarized as follows:

- A novel simple and efficient Active Learning criterion that performs well for the Novel Class Retrieval task.

- An evaluation metric for the retrieval task that considers the spatial layout of the retrieved class.

- An evaluation on long-tailed datasets in order to disregard the class size constraint during the Active Learning process.

- A Novel Class Retrieval framework that allows different types of descriptors, AL strategies and metrics, that could be useful to reproduce our work and for future research work.

## 2 Related Work

In this section, we review existing literature relevant to our problem.

**Novel Class Retrieval:** NCR is defined as the task of iteratively creating a novel class of samples, unknown beforehand, based on the pattern contained in an initial query. In our case, the query is provided by the user. Different from Novel Class Discovery Troisemaine et al. (2023) that aims at discovering all novel classes from an unlabeled pool of data, NCR only extracts one class at a time, that fits a certain pattern. NCR also relates to Few-Shot Learning (FSL) Song et al. (2023b), that tries to learn from a very limited number of labeled examples. FSL tries to infer which already seen class the newly available unlabeled samples belong to. NCR is different in the way that it tries to retrieve a whole set of samples of the same category as the query based on feedback, rather than simply affect it to a class based on similarity alone. Another machine learning task that is closer to NCR is Content-Based Image Retrieval (CBIR) Long et al. (2003); Hameed et al. (2021); Srivastava et al. (2023). CBIR belongs to a larger scope, that is the Image Retrieval scope, while relying on an initial content-carrying query. The main difference between CBIR (and Image Retrieval in general) and NCR is related to the complexity of the expected results, which is the fine-grained aspect of CBIR, as the retrieved images are most of the time similar replicas of the input query. In the NCR context, retrieved images do not follow this constraint: the goal is to rather create a class, with different dispositions, backgrounds and instances of the concept of interest. Despite these differences, CBIR presents a good proxy for NCR, and the different tools to solve the former task can be used for the latter.

CBIR, and by extension NCR, is comprised of two steps: 1) the feature extraction step to describe the images, and 2) the retrieval step. For the feature extraction step, initial works primarily used shallow handcrafted features like color, texture, shape, and key-point descriptors such as SIFT Lowe (2004); Ke & Sukthankar (2004), SURF Bay et al. (2006), Fisher Vectors Perronnin et al. (2010), etc. However, deep global features Chen et al. (2022) extracted from Convolutional Neural Networks Babenko & Lempitsky (2015); Tolias et al. (2015); Tzelepi & Tefas (2016) or Vision Transformers El-Nouby et al. (2021); Song et al. (2023a) now present a better description capacity for the retrieval. For the retrieval step, similar images to the query are extracted to be presented to the user.

Of special importance to our problem are the works that employ iterative schemes where a human-in-the-loop provides Relevance Feedback (RF) during this step, as NCR is by definition an iterative process based on

human interaction. During such schemes, a classifier is iteratively refined based on the user's feedback, which is usually in the form of annotations. The question is how to choose the samples for which the feedback is informative. Traditional RF tends to select the samples closest to the initial query, in terms of similarity, or the classifier's prediction Ngo et al. (2016). This form of feedback, called *passive feedback*, suffers from the lack of informativeness with regard to the classifier training. However, the passive feedback criterion has the advantage of rapidly finding *relevant* samples to the class-of-interest, thus improving the user's satisfaction during the process. Our work makes use of this advantage, while introducing enough informativeness to the classifier. Another subset of RF methods uses AL to answer this question Tong & Chang (2001); Ngo et al. (2016).

**Active Learning:** AL Settles (2009) is a ML tool that aims at selecting the most *useful/informative* samples from an unlabeled dataset to be labeled by an oracle. AL can be used both for classification and retrieval purposes. When only a limited annotation budget is available, AL strategies show better performances than random selection. Such strategies can be categorized through their definition of the *informativeness* of a sample. This *informativeness* can be seen as: uncertainty, diversity, representativeness, committee disagreement, model change, error reduction, variance reduction, etc.

Because of their simplicity and effectiveness, uncertainty-based strategies are of central importance to our work. Such strategies exploit the uncertainty of the model prediction to select the queried samples. Least confidence sampling Lewis & Gale (1994); Lewis & Catlett (1994) uses the probability of the most probable class, and chooses the samples near the classifier margin, i.e. with a probability close to 0.5. Margin sampling Scheffer et al. (2001); Tong & Chang (2001) computes the difference between the probabilities of the first and second most probable classes, so that the samples with the lowest difference are selected. Entropy sampling Shannon (1948); Joshi et al. (2009) uses entropy as an uncertainty measure, and selects the samples with the highest entropies. These criteria are often used alone, or coupled with other diversity methods Brinker (2003); Xu et al. (2007), which is computationally expensive in our interactive setting. Our work uses the least confidence sampling approach, but differs from it by tweaking the set of selected samples to take into account the retrieval component of our NCR problem. Some other AL works Kothawade et al. (2021); Sener & Savarese (2017) mainly rely on the representativeness of the unlabeled set by the selected samples. They try to find the most representative unlabeled subset, that allows for training a classifier whose performance can achieve that of a classifier trained on the whole dataset. However, the selected samples in the NCR case should rather cover the class-of-interest, which we have no prior about. This makes such strategies inadequate to our setup.

A recent line of work incorporates deep learning into the AL domain. Works such as BALD Gal et al. (2017), CEAL Wang et al. (2016) and Core-Set Sener & Savarese (2017) finetune a neural network rather than training a lightweight classifier. Such strategies require a relatively large annotation budget in order to modify the network weights. At each AL iteration, this large batch of annotated samples is included in the training data, which results in important training times. This is unfeasible in our iterative RF context, as wait times, and by extension training times, should be limited to seconds, and annotation budgets should not exceed tens of images. Aggarwal et al. (2022) shows that, in settings similar to ours, using a linear lightweight classifier shows better results than the finetuning of a neural network. The chosen budget remains larger than ours, around 200 samples to annotate, but we adopt a similar strategy. We also show that our criterion outperforms the formulated method in this work, while being conceptually simpler.

## 3   Methodology

In this section, we first formulate our setup before presenting our novel **Positive-First Most Ambiguous** Active Learning selection criterion.

### 3.1 Problem Formulation

Our initial data is a set of $N$ unlabeled images $\{I_i\}_{i \in [1,N]}$, encoded using a pretrained neural network $\Phi$ to obtain the initial dataset $D = \{x_i\}_{i \in [1,N]}$:

$$x_i = \Phi(I_i) \in \mathbb{R}^d \tag{1}$$

To retrieve the class-of-interest, the user provides a first query containing $N_p$ positive images, i.e. that contain the desired concept, and $N_n$ negative images, i.e. different than the desired concept. In this study we consider very small values of $N_p$ and $N_n$ ($N_p = 1$ and $N_n = 5$ in our experiments) corresponding to a very limited annotation budget scenario. The provided query is used to initiate a labeled training set $D_l$, where positive images are labeled 1 and negative images are labeled 0.

The learning and retrieval loop is then performed for $T$ iterations. In our experiments, we consider values of $T$ up to $T = 25$ iterations. Each iteration $1 \leq t \leq T$ is conducted as follows:

1. A classifier $f$ is trained on $D_l$ to recognize the class-of-interest, and classify each image as positive, i.e. in the class-of-interest, or negative, i.e. outside the class-of-interest:

$$f : x_i \in \mathbb{R}^d \mapsto [0,1] \tag{2}$$

2. An AL strategy is used to score each sample in $D \smallsetminus D_l$. The scores are ranked in decreasing order.

3. Given an annotation budget $b$, the top $b$ samples are selected based on their given scores. The selected set $S_t$ contains the samples with the highest scores:

$$S_t = \{x_{i_s}\}_{i_s \in [1,b]} \tag{3}$$

   where $i_s \in [1,N]$ and $i \to i_s$ gives the rank of the $i^{th}$ sample w.r.t. the decreasing order of the score.

4. The selected samples are annotated by the user and added to the training data:

$$D_l \leftarrow D_l \cup S_t \tag{4}$$

As the loop is performed in a interactive manner, where the user labels the selected samples, not only the wait time of the user before the next set of selected samples should be realistic, of the order of a few seconds, but also, the amount of selected samples should be easily handled by the user. These constraints impose the use of a light model for the classifier, as we are in interactive low-annotation budget setting. Like the work of Aggarwal et al. (2022), we use a *linear SVM*. We, however use a much smaller budget $b = 10$, where Aggarwal et al. (2022) uses $b = 200$.

### 3.2 Positive-First Most Ambiguous AL Strategy

As mentioned before, our selection criterion should not only select samples that train a good classifier, but also ensure that at each iteration, a large number of diverse positive images covering the class-of-interest is added. Two criteria arise:

- An uncertainty criterion that provides maximum informativeness to the classifier. Because of the short wait times between iterations, we use the simplest uncertainty criterion which is least confidence, called *MA* for *Most Ambiguous*. The score for each sample $x_i$ can be written as:

$$MA(x_i) = 1 - |0.5 - f(x_i)| \tag{5}$$

  Also because of the short wait times, we don't couple *MA* with any diversity or representativeness measure. This method guarantees a performant classifier but fails to respect the rapid class filling requirement. This requirement is of crucial importance, since the human user may decide to stop before finishing the process. As a result, relevant samples should be presented since the start.

- A criterion that ensures the selection of positive samples. This strategy is called *MP* for *Most Positive*. The score following this strategy is:

$$MP(x_i) = f(x_i) \tag{6}$$

This method allows for fast retrieval, but does not provide enough information to the classifier, as the selected samples live in subspaces already known by the classifier.

Fig. 1 (top and middle) clearly showcases the advantages and inconvenient of these two strategies: *MA* provides very diverse and informative samples, but they are mostly negatives, whereas *MP* retrieves positive samples relevant to the class-of-interest, but very similar, therefore lacking novel information.

Our *Positive-First Most Ambiguous* (*PF-MA*) strategy relies on the advantages of each the criteria. The idea is that the selected samples should still be ambiguous to bring enough information, but should also have more chance to be positive to fill the class-of-interest. Based on these considerations, we formulate the score of the novel criterion as follows:

$$\begin{aligned} PF\text{-}MA(x_i) = MA(x_i) &\times \mathbb{1}_{(f(x_i) \geq 0.5)} \\ + MP(x_i) &\times \mathbb{1}_{(f(x_i) < 0.5)} \end{aligned} \tag{7}$$

In a more detailed manner:

$$\begin{aligned} PF\text{-}MA(x_i) = (1 - |0.5 - f(x_i)|) &\times \mathbb{1}_{(f(x_i) \geq 0.5)} \\ + f(x_i) &\times \mathbb{1}_{(f(x_i) < 0.5)} \end{aligned} \tag{8}$$

More intuitively, *PF-MA* distinguishes between the positive most ambiguous samples, and the negative most ambiguous samples. The positive most ambiguous samples are chosen first, then in case there are not enough samples to fill the budget $b$, it selects from the negative ones.

### 3.3 kNN Acceleration

In the context of large scale databases and a human-in-the-loop search, scoring each sample in $D \setminus D_l$ is computationally intractable while maintaining acceptable wait times. The idea is to reduce the set of candidates $C_a$ to select the samples to label from, and thus that need to undergo scoring. We choose the new set of candidates based on similarity search, by choosing the samples closest to the already selected ones. At each step $t$, the positive and negative selected samples are all stored in $D_l$, and the set of chosen candidates are the nearest neighbors of the elements of $D_l$:

$$C_a = \bigcup_{q \in D_l} \mathcal{NN}(q, k) \tag{9}$$

where $\mathcal{NN}(\cdot, k)$ is the set of the $k$ nearest neighbors. We use $k = 200$ in our experiments.

Typically, this kNN acceleration is most effective when coupled with hashing methods and index structures (Joly & Buisson (2008); Silpa-Anan & Hartley (2008); Jegou et al. (2010); Lv et al. (2017)) to speed up the neighbors search. However, in this paper we are mainly interested in studying the effect of using knn rather than the whole dataset when selecting samples. We have therefore implemented only an exhaustive search for knn.

## 4 Experimental Framework

### 4.1 Data and Experimental Setup

**Datasets:** Real world datasets frequently suffer from skewed class proportions, also known as class imbalance. This often results in classifiers having better performances regarding majority classes, whereas minority classes are overlooked. The problem is that synthetic datasets often used for different tasks do not

showcase this problem. The classes in such datasets are rather well-balanced. In order to mimic real world imbalance, new datasets have been created. These datasets have long-tailed class sizes distributions, where a few classes account for the majority of the dataset, and a larger share of classes represent a very small number of samples. We make use of such datasets as their imbalance presents a great proxy for the unknown size of the class-of-the interest. In other words, the user might be interested in creating a class with rather common pattern present in a relatively large share of the data, or a more unique pattern that only very few images contain. We use the datasets **Cifar100-LT** Cao et al. (2019) with an imbalance factor of $IF = 50$, **ImageNet-LT** Liu et al. (2019) with $IF = 256$ and **PlantNet300K** Garcin et al. (2021) with $IF = 500$, where the imbalance factor is defined as:

$$IF = \frac{\text{number of images of majority class}}{\text{number of images of minority class}}$$

For comparison and discussion purposes, we also evaluate on balanced datasets including **Cifar100** Krizhevsky et al. (2009) and **ImageNet** Deng et al. (2009).

**Descriptor models:** We use two pre-trained models to describe our images, both of them based on the *ViT-L14* Dosovitskiy et al. (2020) architecture, but trained via two different frameworks: **CLIP** Radford et al. (2021) and **DINOv2** Oquab et al. (2023), as they showcase great performances for other tasks. The main difference between both frameworks is that DinoV2 model is pre-trained using self-supervision (on images only), whereas CLIP is pre-trained using contrastive learning on images and associated textual descriptions (language-supervised). In addition, CLIP could allow for the future use of textual queries.

**Initial query:** For the retrieval, we randomly select $Q$ queries per class, where each query contains $N_p$ positive image and $N_n$ negative images. Generally, $N_p \leq N_n$ as the user can initially provide very few positive images, compared to negative ones. We choose $Q = 10$ to ensure diversity within the initial query, and thus generalization. We set $N_p = 1$ and $N_n = 5$.

**Baselines:** We compare our criterion to straightforward baselines as well as more advanced criteria from the state-of-the-art:

- *MA*: Most Ambiguous.

- *MP*: Most Positive.

- *DAL*: Discriminative Active Learning Gissin & Shalev-Shwartz (2019). *DAL* chooses the samples that are more similar to the unlabeled data than to the labeled data, using a labeled vs. unlabeled classifier.

- *ALAMP* Aggarwal et al. (2022). We choose *ALAMP* because of its close settings to our problem, i.e. a small budget and a linear classifier. In brief, *ALAMP* assigns scores to samples at iteration $t$ as follows:
$$ALAMP(x_i) = \frac{marg_{t-1}(x_i) - marg_t(x_i)}{marg_{t-1}(x_i) + marg_t(x_i)} \tag{10}$$

  where $marg_t(x_i)$ is the margin sampling score. The margin sampling score is defined as the difference between between the probabilities of the top-2 predicted classes. The larger the margin score, the further the probabilities assigned, and the more certain the sample is. As a result, *ALAMP* prioritizes samples that switched from high certainty to high uncertainty. When the difference in uncerainty is the same, the denominator prioritizes samples with overall low certainty.

We also combine the uncertainty strategies with diversity methods. We choose straightforward diversity terms, to keep the computation times low:

- *\*-S* (for step): where * ∈ {*MA*, *MP*}. The samples are ensured to be diverse by selecting one sample each $S = 5$ samples from the set of ordered samples.

- *\*-D* (for distance): where $* \in \{MA, MP\}$. A larger number $B = 50$ of samples is selected using the uncertainty criterion to form a pool. Then the $b$ samples are selected from this pool by iteratively sampling the farthest sample from the already selected ones.

**Annotation budget:** Aggarwal et al. (2022) clearly mentions that small budgets (200 annotations) present a challenge for AL. In our case, the annotations are conducted by the user, making it impossible to annotate hundreds of samples. Therefore, we further restrict the annotation budget to merely $b = 10$ samples at each iteration, for up to $T = 25$ iterations, which is even more challenging.

**kNN acceleration:** Unless mentioned otherwise, the kNN acceleration is used only with ImageNet (the biggest dataset). We conduct an exhaustive selection for the other datasets.

### 4.2 Metrics

To evaluate the performance of our novel AL strategy for Novel Class Retrieval, we use two types of metrics: metrics for the classifier performance, and metrics for the retrieval performance. For the classifier performance, we use a traditional **f1-score** on a held-out test set.

For the retrieval performance, a first metric is the proportion of **returned positives**. For a class-of-interest $C$ of size $k_C$, let $Pt$ be the set of positive samples returned up to the iteration $t$:

$$pos_t^C = \frac{\#P_t}{k_C} \tag{11}$$

However, the number of positives returned does not really reflect how well the retrieval is. One problem of this metric is that the samples returned can be close to each other, which creates a less diverse class. For example, the user may be interested in the same object or pattern, but with different backgrounds, states, or positions. Most alternative retrieval metrics also focus on the number of returned positives (e.g. mean average precision). A class of metrics that compute diversity of a set of samples rely on the pairwise distances between the samples. However in the case of differently distributed classes like our case (i.e. some classes may be concentrated, some may be spread out), a low distance may not always mean a lack of diversity. Another class, like the Vendi score Friedman & Dieng (2022), is based on the entropy of the variances of the selected set. In other words, the more spread out across different directions a set is, the better its diversity, and the more concentrated, say across one dimension, the less its score. These metrics are not adequate to compute the diversity of a subset within a reference set, no matter the structure of this reference set. In our case, we mainly care that the model queries the user on all regions of the class-of-interest, regardless of its topology. Thus, the purpose of our new metric is to cover the maximum feature space regions that belong to our class. If the class-of-interest is scattered, for instance, we would rather have our selected samples also scattered rather than confined to a sub-cluster of the class. This ensures diversified samples. To this end, we propose to use a **class coverage** metric $cov_t^C$. Therefore, we run a *K-means* with $K = 32$ clusters $\{CLS_i\}_{i \in [1,K]}$ on the class $C$ samples (this is done offline separately for each class of the whole dataset). Then, to evaluate a particular retrieval iteration, each returned positive sample is assigned to a cluster. The class coverage is defined as the proportion of clusters that are assigned at least one positive sample:

$$cov_t^C = \frac{\#\{CLS_i | \exists x \in P_t, x \in CLS_i\}}{K} \tag{12}$$

For a more stable coverage estimation, we actually perform 10 *K-means* clusterings, then average the results. For classes with less that $K$ samples, each sample is considered as a cluster.

### 4.3 Our Framework

We implemented a framework to experiment with and evaluate Novel Class Retrieval. Our framework includes different datasets, feature extractors, Active Learning strategies and evaluation metrics. It also allows for the easy addition of new elements for each of these categories. Its use is quite forward as it suffices to pick a dataset, an extractor and an AL strategy. Different configuration files enable the modification of the

hyper-parameters. For each retrieval experiment, the framework provides a full summary of the experiment, as well as computes the different metrics during each retrieval iteration for a detailed evaluation.

## 5 Experimental Results

### 5.1 Performance Factors: Search Iterations and Class Size

We start by presenting and analyzing the results with regard to the number of search iterations and the class-of-interest size. We base our analysis on the dataset ImageNet-LT, using DINOv2 as a feature extractor.

**Varying the search iteration:** Here we present the retrieval results per iteration. The results are averaged for all classes, and presented in fig. 2. As mentioned before, *MA* has the best classifier performance, but does not do well when it comes to filling the class-of-interest. On the other hand, *MP* has a reversed behavior. *ALAMP* seems to behave closely to *MA*, as it also picks informative samples, with no regard to their relevance to the class-of-interest. Our method *PF-MA* presents the best compromise between these two tasks, while showcasing a great coverage that surpasses all methods. Our method has the best coverage results, as it first selects samples near the boundary but that are positive. Diversity is encouraged as near duplicates of the initial query are not selected to start with. During the first iterations, the samples are mainly located near the boundary, so far from the region where the positive query resides, which allows our strategy to cover and explore this new region. Moving forward, as the boundary region is now known, *PF-MA* will select samples in between, i.e. not very ambiguous, but closer to positive, which unlocks new regions in the feature space of our class. This helps span different regions of the class-of-interest. We showcase through our experiments that the mean score of the selected samples starts at 0.5, and continuously increases to achieve 0.75 during the later iterations.

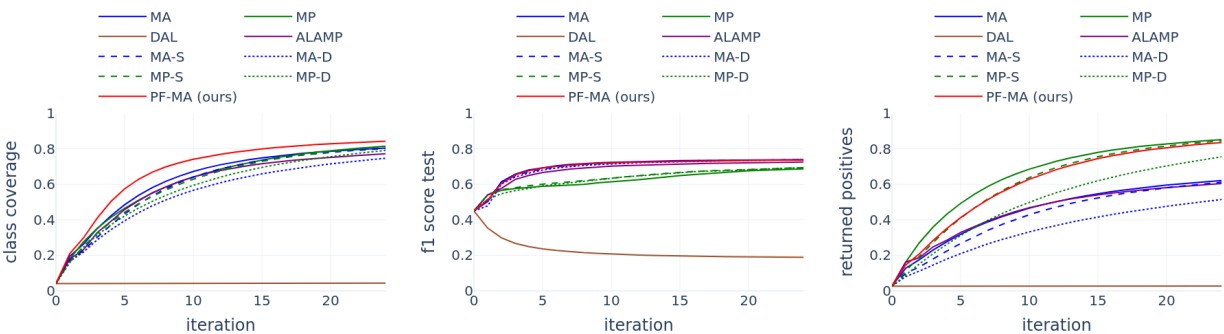

Figure 2: Search results per iteration. Class coverage $cov_t$ (left). F1-score (middle). Returned positives $pos_t$(right).

**Varying the class size:** We analyse the effect of the class-of-interest size on the retrieval performances. We present the results for iterations 5, 15 and 25 in fig. 3, 4 and 5. *MP* performs well for all metrics for small classes. As the size increases, however, *MP* performances decrease for the f1-score and the class coverage. This demonstrates the lack of diversity of the samples selected by this method. The class-of-interest is not well covered and the classifier does not perform well on the held-out test set. For larger classes, *MA* shows better results, but always struggles to return a competitive number of positives, which doesn't serve the retrieval purpose. If we had to choose between *MA* and *MP*, the user would need to know the size of the class-of-interest in advance, and pick accordingly. However, this information is usually impossible to know beforehand. Our method *PF-MA* maintains the best performing strategy, no matter the class size, or the number of search iterations. *PF-MA* can be applied without any additional prior knowledge regarding the size. Last but not least, another important advantage of *PF-MA* is that it presents the best coverage performances almost everywhere. Also note how *DAL* have very low results for all configurations. In the original work, the AL loop is initialized with 5000 samples on which the first is trained, which holds way more information than we use in our setting.

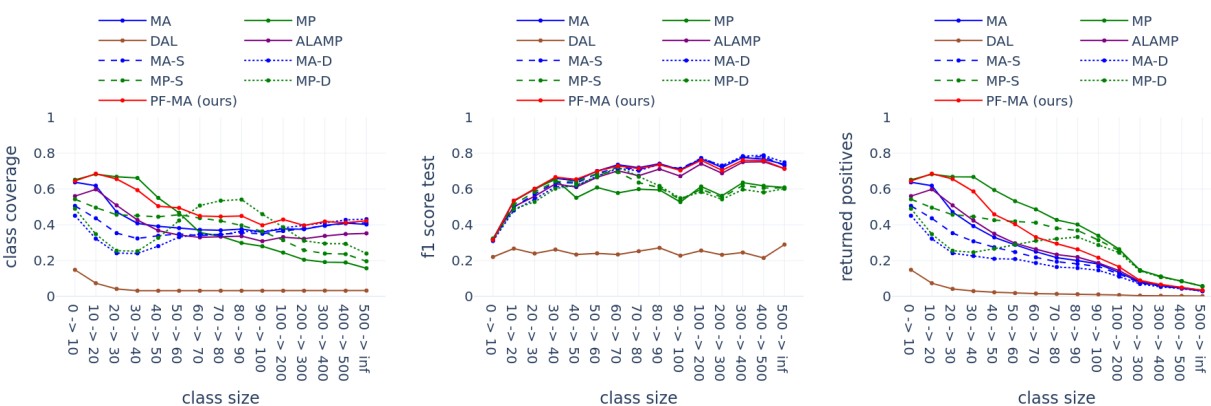

Figure 3: Search results at iteration 5 per category of class size. Class coverage $cov_5$ (left). F1-score (middle). Returned positives $pos_5$ (right).

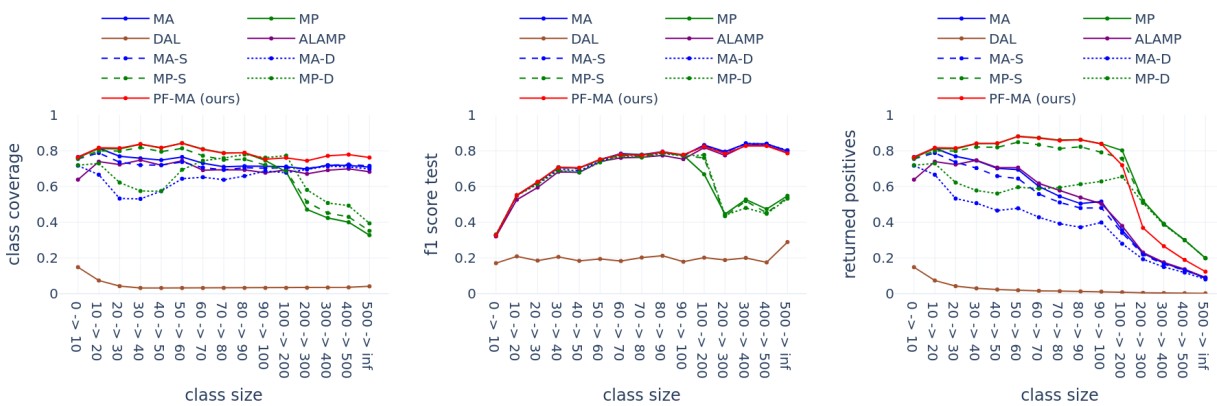

Figure 4: Search results at iteration 15 per category of class size. Class coverage $cov_{15}$ (left). F1-score (middle). Returned positives $pos_{15}$ (right).

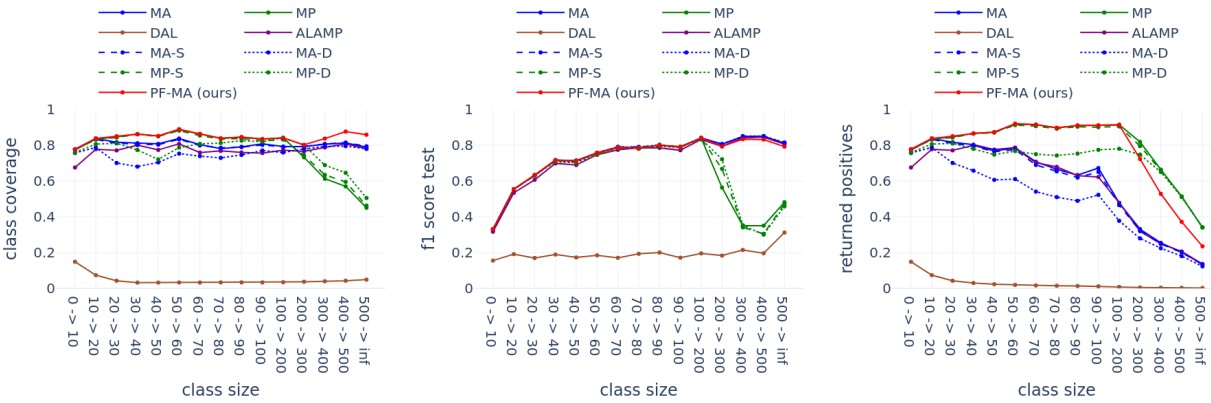

Figure 5: Search results at iteration 25 per category of class size. Class coverage $cov_{25}$ (left). F1-score (middle). Returned positives $pos_{25}$ (right).

In the next sections, we mainly rely on the class coverage metric for the evaluation. On one hand, the f1-score only measures how good our classifier is, which is not a proxy for the retrieval effiency, as we have noted with *MA*. On the other hand, the proportion of returned positives does not provide any idea about the diversity of the retrived samples. We actually believe that the class coverage is the most appropriate metric to capture both aspects of the retrieval performance: recall and diversity.

Table 1: Class coverage scores at iterations 5, 15 and 25 for different AL methods, imbalanced datasets (Cifar100-LT, ImageNet-LT and PlantNet300K) and descriptors.

| metric | method | Cifar100-LT | | ImageNet-LT | | PlantNet300K | |
|---|---|---|---|---|---|---|---|
| | | CLIP | DINOv2 | CLIP | DINOv2 | CLIP | DINOv2 |
| $cov_5$ | *MA* | 0.402 | 0.502 | 0.351 | 0.423 | 0.203 | 0.282 |
| | *MP* | 0.41 | 0.454 | **0.368** | 0.396 | **0.212** | 0.29 |
| | *DAL* | 0.04 | 0.039 | 0.043 | 0.043 | 0.051 | 0.05 |
| | *ALAMP* | 0.349 | 0.471 | 0.289 | 0.387 | 0.148 | 0.235 |
| | *MA-S* | 0.363 | 0.442 | 0.321 | 0.377 | 0.188 | 0.257 |
| | *MA-D* | 0.388 | 0.407 | 0.327 | 0.355 | 0.193 | 0.284 |
| | *MP-S* | 0.381 | 0.435 | 0.34 | 0.381 | 0.193 | 0.27 |
| | *MP-D* | 0.409 | 0.429 | 0.347 | 0.378 | 0.198 | 0.294 |
| | *PF-MA (ours)* | **0.411** | **0.56** | 0.36 | **0.493** | 0.204 | **0.298** |
| $cov_{15}$ | *MA* | 0.801 | 0.838 | 0.76 | 0.736 | 0.487 | 0.59 |
| | *MP* | 0.773 | 0.785 | 0.713 | 0.688 | 0.456 | 0.537 |
| | *DAL* | 0.072 | 0.066 | 0.045 | 0.044 | 0.053 | 0.051 |
| | *ALAMP* | 0.704 | 0.83 | 0.638 | 0.7 | 0.318 | 0.514 |
| | *MA-S* | 0.775 | 0.821 | 0.735 | 0.722 | 0.465 | 0.57 |
| | *MA-D* | 0.815 | 0.74 | 0.734 | 0.653 | 0.48 | 0.588 |
| | *MP-S* | 0.756 | 0.785 | 0.694 | 0.681 | 0.435 | 0.523 |
| | *MP-D* | 0.774 | 0.736 | 0.697 | 0.647 | 0.447 | 0.542 |
| | *PF-MA (ours)* | **0.824** | **0.915** | **0.773** | **0.79** | **0.488** | **0.604** |
| $cov_{25}$ | *MA* | 0.89 | 0.899 | 0.852 | 0.804 | **0.596** | 0.678 |
| | *MP* | 0.875 | 0.877 | 0.821 | 0.776 | 0.565 | 0.628 |
| | *DAL* | 0.101 | 0.096 | 0.049 | 0.046 | 0.055 | 0.054 |
| | *ALAMP* | 0.811 | 0.892 | 0.744 | 0.771 | 0.406 | 0.605 |
| | *MA-S* | 0.88 | 0.895 | 0.844 | 0.802 | 0.583 | 0.671 |
| | *MA-D* | 0.903 | 0.84 | 0.842 | 0.751 | 0.595 | 0.679 |
| | *MP-S* | 0.869 | 0.882 | 0.815 | 0.777 | 0.552 | 0.625 |
| | *MP-D* | 0.881 | 0.86 | 0.816 | 0.76 | 0.563 | 0.639 |
| | *PF-MA (ours)* | **0.908** | **0.954** | **0.861** | **0.844** | **0.596** | **0.684** |

## 5.2 Generalization on Other Databases and Descriptors

In this section, we check if our previous results hold in different settings. We report the class coverage scores at iterations 5, 15 and 25, for both long-tailed and balanced datasets in tab. 1 and 2. We do not perform *ALAMP* and the diversity combined strategies on ImageNet due to computation times.

For the imbalanced datasets, our method mostly presents the best results for all datasets and both feature extractors. Overall, our criterion improves the retrieval results. In almost all cases, the coverage score is significantly better. In a few cases only, during the first iterations of CLIP, *PF-MA* shows a slight decrease in performance. We notice overall better results when using DINOv2 for feature extraction. We also note larger gains with Cifar100-LT and ImageNet-LT.

For the balanced datasets, we underline the second-best method in tab 2. Although our method was designed for real-word cases where the size of the class-of-interest varies, we notice that it remains second best in the worst case, where it is outperformed by *MA* combined with distance-based diversity. We explain the good

Table 2: Class coverage scores at iterations 5, 15 and 25 for different AL methods, balanced datasets (Cifar100 and ImageNet) and descriptors.

| metric | method | Cifar100 | | ImageNet | |
|---|---|---|---|---|---|
| | | CLIP | DINOv2 | CLIP | DINOv2 |
| $cov_5$ | MA | 0.329 | 0.424 | 0.247 | 0.226 |
| | MP | 0.271 | 0.182 | 0.207 | 0.131 |
| | DAL | 0.034 | 0.034 | 0.072 | 0.056 |
| | ALAMP | 0.313 | 0.379 | - | - |
| | MA-S | 0.345 | 0.431 | 0.258 | 0.23 |
| | MA-D | **0.396** | **0.464** | 0.293 | **0.276** |
| | MP-S | 0.31 | 0.224 | 0.236 | 0.151 |
| | MP-D | 0.381 | 0.28 | **0.294** | 0.19 |
| | PF-MA (ours) | 0.331 | 0.432 | 0.265 | 0.274 |
| $cov_{15}$ | MA | 0.731 | 0.79 | 0.595 | 0.565 |
| | MP | 0.539 | 0.381 | 0.416 | 0.248 |
| | DAL | 0.057 | 0.052 | 0.087 | 0.066 |
| | ALAMP | 0.712 | 0.765 | - | - |
| | MA-S | 0.732 | 0.794 | 0.597 | 0.564 |
| | MA-D | **0.823** | 0.794 | **0.671** | **0.64** |
| | MP-S | 0.555 | 0.413 | 0.429 | 0.259 |
| | MP-D | 0.627 | 0.472 | 0.506 | 0.309 |
| | PF-MA (ours) | 0.838 | **0.838** | 0.607 | 0.599 |
| $cov_{25}$ | MA | 0.856 | 0.882 | 0.749 | 0.715 |
| | MP | 0.696 | 0.559 | 0.549 | 0.332 |
| | DAL | 0.083 | 0.078 | 0.094 | 0.071 |
| | ALAMP | 0.85 | 0.867 | - | - |
| | MA-S | 0.852 | 0.883 | 0.749 | 0.715 |
| | MA-D | **0.912** | 0.878 | **0.813** | **0.763** |
| | MP-S | 0.704 | 0.583 | 0.555 | 0.338 |
| | MP-D | 0.756 | 0.634 | 0.618 | 0.385 |
| | PF-MA (ours) | 0.871 | **0.935** | 0.763 | 0.742 |

performance of diversity strategies for these two datasets by the large class sizes (more than 600 images per class). Beyond the unrealistic setting of balanced datasets, another problem is the computational cost of the diversity and the iterative choice of the next sample to select. Due to these costs, diversity-based strategies are unable to scale. Our method, however, offers very acceptable results even in the balanced case, with no additional costs, which provides a great tradeoff.

## 5.3 Effect of kNN Acceleration

Here, we ablate the kNN acceleration to study its effect. We conduct the experiments on the smaller datasets, as the computation time for an exhaustive selection on ImageNet is untractable. We present the class coverage results for our method at iterations 5, 15 and 25 in tab. 3. The degradation of the results is minimal, and the acceleration is sometimes even beneficial to the retrieval performance, especially during the first iterations. We hypothesize that the kNN acceleration improves the results at the beginning of the retrieval due to the exploration of the region of the feature space surrounding the query, rather than elsewhere, which allows the SVM a more comprehensive knowledge regarding this area. The kNN acceleration thus allows a very high acceleration of the search loop, without loss of the retrieval quality. The gain is mainly during the inference and sort times. For an exhaustive selection, the time complexity of the inference is $\mathcal{O}(N)$ where $N$ is the size of the unlabeled dataset. For an accelerated selection, the complexity is rather $\mathcal{O}(k)$ where $k = 200$ in our case. In addition, the sort is performed on much smaller sets of samples, that don't exceed a size of $(N_p + N_n + (T - 1) \times b) \times k$ vs. millions of samples for larger datasets. However, computing

the kNNs themselves requires an additional cost. The kNN exhaustive search is $\mathcal{O}(N)$, which cancels the acceleration gain of the selection phase. With an efficient indexing structure such as multi-probe locality sensitive hashing Joly & Buisson (2008); Lv et al. (2017), randomized kd-trees Silpa-Anan & Hartley (2008) or PQ-code Jegou et al. (2010), the complexity is typically $\mathcal{O}(N^\gamma)$ where $\gamma < 1$ is the compression coefficient, i.e. sublinear in the dataset size. In our framework, we don't use indexing structures, but we bypass the cost by pre-computing the kNNs for all dataset elements beforehand. The results are then stored and easily accessed whenever needed. This way, the kNN computation cost is not included in the retrieval loop. This cost can be further lowered with efficient approximation packages such as FAISS Danopoulos et al. (2019).

Table 3: Class coverage scores at iterations 5, 15 and 25, using *PF-MA*, with and without kNN acceleration.

| metric | kNN acceleration | ImageNet-LT | | PlantNet300K | |
|---|---|---|---|---|---|
| | | CLIP | DINOv2 | CLIP | DINOv2 |
| $cov_5$ | yes | **0.362** | **0.494** | **0.216** | **0.313** |
| | no | 0.36 | 0.493 | 0.204 | 0.298 |
| $cov_{15}$ | yes | 0.741 | **0.794** | 0.47 | 0.594 |
| | no | **0.773** | 0.79 | **0.488** | **0.604** |
| $cov_{25}$ | yes | 0.829 | **0.847** | 0.57 | 0.674 |
| | no | **0.861** | 0.844 | **0.596** | **0.684** |

## 6 Conclusion

We present a novel Active Learning criterion for Novel Class Retrieval through Relevance Feedback. Our *PF-MA* criterion is based on both samples ambiguity and their class membership, in order to gain enough informativeness while fulfilling the retrieval task. Unlike existing strategies that satisfy one objective only at a time, we show good performances across both objectives. We emphasize that while *Most Positive* already works very well on small classes, we achieve good results for all class-of-interest sizes, discarding the need for any prior knowledge. We showcase that our criterion presents satisfying performances with very small annotation budgets, outperforming both classical and novel criteria in this aspect. In addition, our novel class coverage metric allows for a better evaluation of the retrieval performances by allowing to take into account the whole class patterns. To the best of our knowledge, our work is the first to introduce the use of Active Learning strategies for Novel Class Retrieval purposes. Our findings lay the foundation for further works to investigate other relevant aspects, such as the refinement and adaptation of the descriptor models and the retrieval efficiency for all objects sizes.

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
