# OpenReview forum: "Positive-First Most Ambiguous: A Simple yet Efficient Active Learning Criterion for Novel Class Retrieval"
_TMLR — Rejected by TMLR_

### Review · Reviewer_mDkV · 2025-01-29

**Summary Of Contributions:**

This manuscript introduces an active learning strategy, designed for Novel Class Retrieval (NCR), called Positive-First Most Ambiguous (PF-MA). It balances informativeness and relevance by prioritizing ambiguous positive samples. To address the need for diversity in NCR, they utilize class coverage, a new metric that measures how well-retrieved samples represent the diversity of the class of interest. They also incorporate a kNN acceleration technique to improve computational efficiency while maintaining performance.

**Audience:**

Yes

**Broader Impact Concerns:**

This paper does not include a Broader Impact Statement, the author should discuss it from some aspects like dealing with long-tailed and imbalanced data, etc.

**Claims And Evidence:**

Yes

**Requested Changes:**

Please look at weaknesses 2-5. need more experimental analysis about these weaknesses.

**Strengths And Weaknesses:**

Strengths:

1. It introduces a novel class coverage metric for diversity consideration.
2. This method tests on balanced and long-tailed datasets to model its efficiency.

Weaknesses:
1. This method is too straightforward and naive, just combining the most positive and ambiguous samples. It does not adapt dynamically to the learning progress and there is no trade-off between exploitation (retrieving certain positives) and exploration (diversifying retrieval).
2. Considering the model design, it would be affected by noisy data (outliers).
3. In kmeans part, the choice of K (number of clusters) is fixed to 32 in their experiments but does not justify why this specific value was chosen.
4. The experiments only hold for shallow classifiers like linear-svm and do not generalize to deep learning-based retrieval.
5. The experimental results indicate that the proposed method does not consistently outperform all baselines across different settings.

---

> ### Author Response · Authors · 2025-02-02
>
> We first want to thank the reviewer for their time and effort, as well as their helpful insights. In the following, we will address each weakness in order to bring sufficient explanation for our different choices.
>
> 1. We completely agree that the method is straightforward, which is what makes it both intuitive and computationally cheap. This is the main idea of our paper, as the goal is to allow for fast interaction with the user, which is the core of NCR: each iteration should take only a few seconds between training the classifier and scoring the samples, while using limited computing resources to run it on large image and video corpora with hundreds of users on a single server. Other more elaborate methods do not verify this constraint, and the ones we experimented in our paper do not perform better. And to the best of our knowledge, this novel inspired method of the advantges two mentioned strategies has never been used, which makes our method, although straightforward, novel.
> Regarding the dynamical adaptation to the learning progress, we have experienced starting from a MA selection, and smoothly moving to a MP selection, with different selection thresholds in between, but the results showed that PF-MA performs better. In fact, our method inherently adapts through the iterations: As the classifier becomes more knowledgeable about our class, there are fewer ambiguous samples, and the majority of the samples suggested are positive with larger probabilities. In the case that it fails again to correctly classify, ambiguous samples are queried. We have experimentally verified that the mean score of the selected samples during the first iterations is around 0.5, and it continuously increases to achieve 0.75 during the later iterations. Our method, therefore, dynamically adapts to the progress.
> In our case, the exploration-exploitation trade-off lies in the use of the positive ambiguous samples, as we exploit certain positive samples, while exploring the boundary. Adding an explicit hyperparameter to showcase this trade-off requires further finetuning to adapt to each used dataset. In the NCR framework, this is not possible as the application datasets are purely unlabeled. It is actually a strength of our method to not need such a hyperparameter, and experiments showcase that it is generalizable to different datasets, different class sizes and different imbalance factors.
>
> 2. The noisy data in our case are a result of an imperfect user that mislabels the queried samples. In this work, we hypothesize a perfect user. The presence of noisy labels is a topic we may address in future work.
>
> 3. Concerning the choice of k, we have analyzed the effect that different values of k would have on our metric. Larger values of k => many classes will not be clustered as their size is < k, very few number of samples per class, some clusters may be empty. Smaller values of k => easier to hit all clusters, not really a good metric as the scores will be very close.
> We have set a threshold : at least 70% of the classes should be clustered. On ImageNetLT and Cifar100LT => anything lower than k = 32 works. But on PlantNet300K => at most 8 clusters, but this is very low as there are also many larger classes.
> We have checked for 16, 32, 64, but our method still performs best, regardless of k. We can provide the quantitative results if requested by the reviewer. Thus, we chose k = 32 to respect the threshold constraint.

---

> ### Author Response · Authors · 2025-02-04
>
> (continued)
>
> 4. In the NCR case, shallow classifiers that work with few data only, like SVM, are sufficient, more stable, and allow for lower variance of the results. During each iteration, a human user provides the labels on which the classifier is trained, which restricts the number of annotated samples, in our case we chose 10. With only a few tens or even a few hundreds of samples, using a neural network classifier is inadequate and unjustified. The results would vary strongly depending on the weight initialization, and the data is not sufficient to achieve convergence. Thus, deep learning-based retrieval is not adapted. In addition, re-extracting the whole dataset’s features, during the wait time of the user, based on the new network, is unfeasible. We did not develop a general AL procedure, but an AL procedure specifically for NCR.
>
> 5. NCR is employed to explore real and raw data, and this data is almost surely never balanced. The best datasets to mimic such data are imbalanced datasets, which is why a large part of our experiments are conducted on the latter. This is the core of our work, as we want our method to perform best in these settings. The experiences on the balanced datasets are mainly to showcase that PF-MA also provides acceptable, and sometimes the best, results on processed data. The only methods that sometimes outperform PF-MA are the diversity based ones. However, as mentioned in the article, these methods are computationally expensive. That is to say, in the interactive setting of our research, they are not the most suitable regarding the time constraint. As a result, our method remains the best choice, for all types of datasets, in the interactive, time-constrained, NCR setting for which it was developed.
>
> Concerning the broader impact statement, we have mentioned in the paper that we work on datasets that mimic realistic datasets, and presented our problem as the problem of exploring larger datasets, which is, nowadays, very much needed. Does the reviewer think we should add a paragraph that explicitly explains this ?

---

### Review · Reviewer_vhPy · 2025-01-30

**Summary Of Contributions:**

The authors design and experimentally verify a new active learning based method for the problem of “Novel Class Retrieval” (NCR), roughly corresponding to the setting where a user U wishes to retrieve a certain class of images from some larger database D (e.g. return the set of “dogs with hats” for a database of animals).

The author’s propose the following iterative “human in the loop” framework for solving this problem. The user (who also acts as the oracle in the active learning sense), first seeds the algorithm with a few positive and negative examples of images in their class. The algorithm trains a [0,1]-valued classifier on these (meant to measure probability of being in the class), and uses the classifier combined with some selection criterion to pick a few images to send back to the user. The user then labels these images and returns them to the algorithm, and the process continues.

The authors consider three metrics for measuring the success of a particular instantiation of the above process: the total number of positive samples received, the error of the final classifier, and the `diversity’ of positive examples shown (i.e. the returned images should in some sense “span” the user’s desired class, they should not be too focused in some small cluster in the solution-space).

This final criterion is really the main focus of the work. The authors evaluate this criterion formally by first clustering the user’s class using k-means, then measuring the fraction of clusters that are hit by the algorithm’s returned samples to the user. They call this metric the `coverage’ of the algorithm.

In order to build a procedure that performs well on all metrics, the authors propose the “Positive-First Most-Ambiguous” (PF-MA) criterion for active sample selection by the algorithm, which, given the results of the classifier on the ith iteration, selects the most ambiguous positive examples, or the most ambiguous negative examples if no positively labeled examples exist.

The authors test this criterion using SVM as the classifier across a variety of datasets and against various other selection criterion and baselines.

**Audience:**

Yes

**Claims And Evidence:**

Yes

**Requested Changes:**

I think the introduction of this work and the formalization (or at least discussion) of the NCR problem needs to be re-written before I can recommend acceptance of this paper. As discussed above, the discussion regarding the setting, retrieval, and active learning was very confusing, and it is still unclear to me what exactly the NCR problem is. I recommend starting the paper (at least in second paragraph) with exactly the problem you are trying to solve (there is a large unlabeled dataset, user wants to retrieve a class from it, etc). This will make the rest of the work easier to read. The current focus on labeled vs unlabeled data and its relation to active learning is very confusing without the problem at hand being properly defined.

*Minor comments:*

Novel Class Retrieval is not standard, so first sentence of abstract is pretty unhelpful. Give a short one sentence description of the task first.

What does “represented by a neural network \Phi” mean? How is this relevant?

In the descriptor model paragraph, maybe be more explicit that you are talking about how the images are represented in space (R^d) in order to fit into your described framework? Could be this is very clear to anyone who does experimental ML, but I didn’t follow what this meant until later.

Typos: “susbet”, “availabilty”, “reproduve”, “sort wait times” “untractable -> intractable”, “do no showcase”, “ablate”(?), “important training times”

**Strengths And Weaknesses:**

*Strengths*

The high level problem setup, and especially the notion of "coverage", are very natural. I can imagine many scenarios in which a user wants to do an image search for some type of result and would want to emphasize getting a `variety’ of images in the class. Practically, clustering the image class and looking at how many clusters are hit seems like a natural way to measure this type of retrieval variety.

The PF-MA criterion and general algorithmic framework is simple and easy to execute.

I have no experience in practical machine learning, but the experiments seem fairly thorough (testing e.g. different sizes of positive class, different numbers of iterations, many different data-sets, two feature extraction/representation methods), and exhibit that the simple PF-MA performs well across the board with respect to coverage without sacrificing too much on total number of

**Weaknesses**

I found the introduction and the formal definition of what the “Novel Class Retrieval” problem actually is to be very confusing. For instance, as far as I can tell the main goal in this scenario is really to retrieve as many (and as varied) images in the users class as possible. Why is the classifier’s accuracy at all a relevant measure for this problem? In related work the authors state “NCR is defined as the task of iteratively creating novel classes of samples, unknown beforehand, based on the patterns contained in an initial query.” I have no clue what this means, and I don’t see how it corresponds to the model they set up. Are there multiple classes? Is the class supposed to be changing between iterations?

Along the same lines, why is there so much discussion of unlabeled data and active learning classifiers in the start of the paper? E.g. in general it makes no sense to specifically try to get the active learning algorithm to return only positive examples — why not just learn the classifier using a general good AL procedure, then at the end just return every image that it labels positively? My understanding after reading the rest of the paper is the reason against this is essentially being in the `small sample’ regime, and trying to get many positive examples as quickly as possible, but this was extremely unclear to me in the beginning.

---

> ### Author Response · Authors · 2025-02-02
>
> We first want to thank the reviewer for their time and effort, as well as their helpful insights. We hope that our following explanation will provide a clearer understanding of our paper. Please note that some changes have been made as requested.
>
> Let’s take the example of the class/concept/pattern “dog with hat”. Our user U wants to create a class of images that match this pattern, by retrieving such images from a large database that doesn’t contain this specific label. This is clearly mentioned in the second paragraph of our introduction as requested by the reviewer.
>
> - Why do we talk about unlabeled datasets? In general, larger databases are not annotated, and even for smaller ones that are annotated, the desired concepts are not part of the labels. For example, for animal datasets, typical annotations are “cat”, “dog”, “horse”, etc. It is rather impossible to find the specific annotation “dog with hat”. In this case, the original labels are useless. As a result, it is appropriate to say that our work hypothesizes that our datasets are unlabeled. Also, the larger share of datasets nowadays is unlabeled.
>
> - How to define NCR? In the article, NCR is defined as “the task of iteratively creating a novel class of samples, unknown beforehand, based on the pattern contained in an initial query and a user’s feedback”. In other words, given our initial query with a certain pattern, we want to create a class that contains this exact pattern, by iteratively retrieving matching samples from our dataset. The key word here is iteratively, which means that the knowledge about the class is refined through multiple iterations. There are not multiple classes, only one class that corresponds to the initial query. And there is no change in this class-of-interest through the iterations, only more knowledge about the desired concept. This knowledge is provided by the user that annotates a set of new samples at each iteration.
>
> - How does NCR work? The first step is that the user provides a query, with one positive image (correct pattern), and some negative ones (incorrect pattern). We create a very small labeled dataset from this query, and train a classifier that tries to recognize positive from negative images. It is clear that the size of this labeled dataset is very small, so the classifier will not be doing a very good job in recognizing the class. So the idea is to use this initial classifier knowledge and query the user for even more refined knowledge. NCR does this by suggesting some samples that he wishes the user to label. Which samples to suggest is where Active Learning intervenes. Then the user labels the suggested samples, which are then added to the labeled dataset, the classifier is retrained, and the loop continues.
> Why is the classifier’s performance important? Because it is the classifier that learns to differentiate between the positive class-of-interest and the rest of the dataset, of course we should be able to see if it can do a good job. It is this classifier that tells us if the knowledge we have gathered about our class is sufficient to easily retrieve this class from the dataset or not.
>
> - Why talk about AL? We have seen previously that NCR should suggest a number of samples to be annotated by the user to refine its knowledge about the class-of-interest. AL helps answer the question of which samples will provide the best knowledge if annotated, and this is why we talk a lot about AL. If we return only positive samples, i.e. samples that the classifier already knows are positive, using these samples to retrain the classifier will not bring any new knowledge. AL hypothesizes that ambiguous samples, i.e. those that the classifier is very unsure about because they are complex, are the ones that maximize the knowledge if annotated. But in our specific case, we want the user to not be bored with annotated very complex samples, without seeing any samples that belong to his class. Hence the new method that verifies both constraints. This is a core notion that should be relayed in the introduction.

---

> ### Author Response · Authors · 2025-02-06
>
> (continued)
>
> - Why not just learn the classifier using a general good AL procedure? This is the core of our work, as we are trying to find this good AL procedure. Traditional AL methods do not respect the diversity constraint and this is the most important point which is mentioned in the introduction (not the small sample regime). We later showcase that they also do not work in our particular settings (limited time, limited budget, etc).
>
> - Why do we talk about the neural network? We just mention that the images are not used in their raw form, but rather encoded into d-dimensional vectors, so that we can use them with our classifier. Generally, shallow classifiers like the one we chose to use, take d-dimensional vectors. We also mention our descriptor model to showcase that we can use whichever neural network we want, as we later tested on 2 networks. We have changed the wording in the paper.
>
> If the reviewer thinks there are still unclear points, we'll be happy to explain them further.

---

### Review · Reviewer_2oAM · 2025-02-03

**Summary Of Contributions:**

This work proposes addresses the problem of Novel Class Retrieval (NCR), where the goal is to retrieve a novel class from an unlabeled pool given a user query ($N_p$ positive images and $N_n$ negative images for a class). This work proposes to address NCR with an active learning inspired solution, in particular, using the proposed PF-MA criterion.

**Audience:**

Yes

**Claims And Evidence:**

No

**Requested Changes:**

- **[Critical to Acceptance]** Improve clarity of writing, and add a figure clearly describing NCR somewhere in the intro.
- **[Critical to Acceptance]** Far more baselines are needed in the experiments section, particularly existing non-AL solutions that do not require iterative training.
- **[Critical to Acceptance]** Justification for diversity metric is needed.

**Strengths And Weaknesses:**

## Strengths
- The paper considers real-world imbalanced datasets in the experiments

## Weaknesses
**Writing Quality Needs Improvement**
- The problem of NCR is not properly described in the introduction and related work. In the second paragraph of the intro, the example provided assumes a weakly annotated retrieval pool which seems to be inconsistent with the setting used in the rest of the paper where the retrieval pool is unlabeled. It is also not clear what "relevance feedback" is in this paragraph, as it is not defined until much later. Similarly in the NCR paragraph of the related work section, it is unclear what the query/relevance feedback are. I would strongly recommend a figure or algorithm block detailing the NCR procedure and rewriting some of the descriptions to improve clarity.
- The motivation of NCR is not clear. The final goal of NCR is not ever clearly stated. Is the goal to train a classifier as in standard active learning? Or is it for user-facing retrieval?
- Active learning in related works is not described accurately. The goal of traditional active learning is not to select **relevant** examples, but rather to select informative samples. Relevance typically implies some notion of similarity, as opposed to uncertainty, diversity, representativeness, committee disagreement, etc.
- Minor typo: unlabaled -> unlabeled in last line of page 3.
- Minor typo: Frawework -> framework in Sec 4.3

**Unclear why AL-style solution is necessary**
- What is the point of proposing a solution that increases $\mathcal{D}_l$ iteratively? This seems to introduce latency, without any clear benefit.
- There are many one-step retrieval solutions that consider diversity, but are not discussed or compared against in the experiments. Maximal Marginal Relevance (MMR) [1] is a classic baseline that considers both diversity and relevance. SIMILAR [2] is a more recent baseline that considers similarity and diversity, and allows for the inclusion of negative samples. Finally, it would be straightforward to use a two-stage strategy that first filters based on similarity (similar to kNN acceleration) and then diversify using any existing diversity-based AL strategy such as CORE-SET [3].

**Weak evaluation metric for coverage**
- First of all, this metric does not penalize imbalanced selections. This metric only considers how many clusters are represented in the set, but does not consider how the samples in the set are distributed across clusters. The latter seems critical in assessing the diversity of a set.
- Second of all, the metric does not consider intracluster diversity.  In otherwords, homogeneity within a cluster is not penalized by the metric.
- It is unclear why the proposed metric should be chosen over existing diversity metric such as Mean Similarity Score or Vendi Score [4], which do not require clustering and measure the dispersion of the features.

**Unclear how proposed criteria encourages diversity**
- It is unclear how the proposed acquisition function encourages the retrieved set to be diverse. This needs to be further explained in the paper.

**References**
- [1] MMR https://www.cs.cmu.edu/~jgc/publication/The_Use_MMR_Diversity_Based_LTMIR_1998.pdf
- [2] SIMILAR https://arxiv.org/pdf/2107.00717
- [3] CORESET https://arxiv.org/abs/1708.00489
- [4] Vendi Score https://arxiv.org/pdf/2210.02410

---

> ### Author Response · Authors · 2025-02-04
>
> We first want to thank the reviewer for their time and effort, as well as their helpful insights. In the following, we will address each weakness in order to bring sufficient explanation for our different choices.
>
> **Writing quality needs improvement**
> - The NCR problem is defined as the iterative creation of a novel unknown class-of-interest based on an initial query by a human user, while exploring a pool of unlabeled data, while relying on the use of human interaction. It is iterative and interactive by nature. The example of the “dog” vs. “cat” dataset is only to showcase that the user is not always interested in the classic labels of the dataset, so it is safe to hypothesize that the dataset is unlabeled, as the novel class has never been defined before. Relevance feedback means the feedback of the user whether the suggested samples are relevant or not, we will make this clearer in the paper. We will also make sure to give more details about the whole NCR process.
> - The final goal of NCR is to simply create a novel class through retrieval, as mentioned “user of a large database aims at creating a class that follows a certain pattern, different than the predefined ones”. The classifier is a mere tool that helps recognize the pattern of the class-of-interest vs. the other patterns.
> - What we meant by relevant samples, ones that are relevant to the training of the classifier, i.e. ones that refine the classifier’s knowledge. As suggested, informative seems like a better wording, we will change that.
>
> **Unclear why AL-style solution is necessary**
> - A user presents an initial query that they want to base their novel class on. The idea is to retrieve samples that fit in this class from a very large unlabeled pool of data. In NCR, the user wants to keep refining their class as long as they can. This is not mere retrieval where the system presents the most relevant “documents”. It is iterative retrieval as mentioned in the definition. The user is presented with samples, he checks whether they are relevant or not, then informs the model. How to refine the retrieval and gather more information about the class if not by using the user’s input? And how do we get the user’s input if we don’t query them on some samples? How to choose these specific samples? AL is the answer. It is quite impossible to create one large labeled dataset, as the human user cannot provide hundreds of annotations at once. NCR is iterative and interactive, and the user may be satisfied during the first iterations, so the whole process may not take more than tens of annotated samples.
> - As mentioned earlier, one-step retrieval solutions are not convenient, as we are dealing with a human user that can only annotate a small number of samples at once, and can stop the process early. The suggested methods are all very computationally expensive, and are developed for large annotation budgets. Also, since NCR is interactive, we want a very simple scoring strategy that doesn’t require large time resources (the user can only wait for a few seconds). Note that MMR seems like a good strategy, it uses similarity to the query (we replace this by the classifier’s score, i.e. MP) and dissimilarity to the other “documents” (the distance-based diversity we test on). The main limitation of MMR is the fixed similarity measure that it uses, and that does not adapt to all class structures (rare vs. dense classes). Training a classifier on the other hand and using its score (MP) provides a more adaptive ranking metric. We have tested on MP-D, which is quite similar in definition to MMR, and PF-MA still performs better, while being less computationally expensive. On the other hand, SIMILAR presents a unified framework for multiple acquisition functions. It uses similarity measures on the the gradients of the loss function w.r.t the inputs. This is not only very computationally expensive as the similarity matrix is huge, but also requires assigning pseudo-labels to hundreds of thousands of inputs based on a classifier trained on a few dozen samples in our setup, which gives a huge room for error. Also, heir notion of diversity is to ensure representativeness of the whole unlabeled data. This is because the question they seek to answer is "what is the best, most representative, subset of data to label to learn a classifier on the dataset ?" which is classic AL. Our setup is very different. The Core-Set approach has the same limitations, as it also tries to find the best subset of data to label to train a classifier. Both SIMILAR and Core-Set know the feature space of the unlabeled data they want to cover, but in NCR, we have no priors about our class.

---

> ### Author Response · Authors · 2025-02-04
>
> (Continued)
>
> **Weak evaluation metric for coverage**
> - Our metric hypothesizes that each dataset class is composed of a set of “subclasses”. We do not want perfectly balanced samples, as long as all the “subclasses” are presented, i.e. all of the class subspace is spanned/covered. In other words, the fact that a high proportion of clusters is included already means that we capture a broad range of variations of the class. The diversity of a set does not mean that all clusters should be equally presented. If this is important, a weighted version of the metric can be developed to account for each cluster’s representation.
> - We do not really want to penalize a homogeneous cluster, as we only care about if that cluster is hit or not. As for heterogeneous clusters, this is why we compute 10 k-means clusterings, and average the results. Clustering with different initializations has a very small chance of grouping the same heterogeneous samples together, so this is how this problem is accounted for.
> - We chose the coverage metric to ensure that we are able to query the user on samples from all class regions. MSS, while effective, only relies on the pairwise distances between the samples of the selected set. It does not take into account the class structure with the feature space. We work with imbalanced datasets, and some classes are very concentrated, while others are very spread out. The mere computation of distances is inadequate, as the structure of our dataset is not uniform. A very spread out class will have a high MSS, even if the selected samples are, for example, from only half of the region of the class. And a very concentrated class will always have a low MSS even if the selected samples cover it very well. On the other hand, Vendi score is based on a similarity function. For the same reason (i.e. some classes are concentrated and some are spread out), similarity would not adequately characterize our classes. As a result, Vendi score inherently gives better scores to more spread out classes. Another area where the Vendi score might fail is when our class is only spread out following few directions, resulting in different eigenvalues and thus a low entropy. Another important critique of these metrics is that they check for the diversity of a set with regard to the whole feature space. Our goal is to rather measure the topological coverage of a subset w.r.t. another set.
>
> **Unclear how proposed criteria encourages diversity**
> - Our strategy selects samples near the boundary but that are positive. Diversity is encouraged as near duplicates of the initial query are not selected to start with. During the first iterations, the samples are mainly located near the boundary, so far from the region where the positive query resides, which allows our strategy to cover and explore this new region. Moving forward, as the boundary region is now known, PFMA will select samples in between, i.e. not very ambiguous, but closer to positive, which unlocks new regions in the feature space. This helps span different regions of the class. We have validated through experiments, as the mean score of the selected samples is around 0.5 during the first iterations, it continuously increases to achieve 0.75 during the last ones.
>
> **Requested changes**
> - We will try to rewrite some parts of our paper for more clarity.
> - NCR is iterative by nature, as the oracle is the human user that want to create a novel class. The iterations helps refine the knowledge about the class, and lighten the annotation load on the user. Non-AL solutions are solely based on similarity the initial query, much like the first iteration of the MP method. We have tried adding diversity through MP-S and MP-D but we can see that it is not sufficient. Testing on non-AL solutions means that we do not use the user's feedback, but our method is specifically developed to use such information, and have been tested against methods that also make use of such knowledge. Based on our previous explanation, does the reviewer still think that such experiments are useful?
> - We have tried to justify our use of the coverage metric, we will include our explanations in the paper.
>
> If the reviewer thinks there are still unclear points, we'll be happy to explain them further.

---

> > ### Comment · Reviewer_2oAM · 2025-02-04
> > **Response to Rebuttal**
> >
> > **One-step solutions are not adaptive**
> > - As far as I now understand it, the user incrementally grows the set of retrieved samples and can specify when to stop when the class is adequately covered. However, in all the experiments the total budget is fixed and therefore comparison with one-step solutions is needed given that this is one of the primary contributions of this work. It may be the case that a good criteria can achieve the same level of coverage in one step as an adaptive AL based method does in multiple steps, thus reducing latency.
> >
> > **MP-D is quite similar in definition to MMR**
> > - This is true, and I had previously overlooked this. Please include a reference to MMR (Carbonell '98) when describing MP-D
> >
> > **Suggested methods are expensive**
> > - It is not clear to me why MP-D/MMR is expensive in this case. If $k$ is the budget size and $n$ is the size of the unlabeled pool, MMR has a complexity of $O(nk^2)$ which is quite feasible for low $k$.
> > - It is mentioned that SIMILAR and Core-Set "know the feature space of the unlabeled data they want to cover", but it seems like this setting also assumes featurized data. It is unclear to me why this is a drawback
> > - Also, SIMILAR does not necessarily require gradient computations and can be used with standard features. Furthermore, the paper assumes a fixed unlabeled pool so computing a similarity matrix is only a one time cost which is incremental (using sparse matrices) given that one needs to featurize the entire pool anyways.
> >
> > **No priors about the class**
> > - The user provides a query which is a class prior? We assume that the user provides a positive example and a few negative example, and we seek to cover the positive example. SIMILAR is used specifically for this purpose i.e we could easily approximate $\text{argmax}_{A} I_f(A; P | N)$ where $P$ denotes the set of positives and $N$ denotes the set of negatives for some information function $f$.
> >
> > **Weak evaluation metric**
> > - The authors' explanation of the weakness of MSS and Vendi score makes sense. However, the reliance on clustering still seems clunky since it requires specifying a $k$ value which is not know a priori.
> > - Perhaps some sort of measure such as MSS(D_l)/MSS(D) or $\sum_{i \in D\D_{l}} \max_{j \in D} sim(i,j) would be more suited for measuring coverage.
> >
> > **PF-MA encourages diversity**
> > - This is still not clear to me. At a given active learning round, an image and its near duplicate will have the same uncertainty score. Thus if a highly uncertain image has a lot of duplicates, there is nothing in the acquisition function preventing all of them from being selected.
> > - It seems like the adaptivity of the selection criteria is the only reason we would expect diversity. However, as stated earlier, a good diversity aware solution (SIMILAR, CORESET, MMR) may be able to achieve good coverage in far fewer steps.

---

> ### Author Response · Authors · 2025-02-04
> **Response to Response to Rebuttal**
>
> **One-step solutions are not adaptive**:
> - As mentioned by the reviewer, the set of retrieved samples, and by extension the labeled set, incrementally grows. This refined knowledge is what helps continuously adapt our classifier score to cover the class. One-step methods will not make use of this refinement, but will only use the initial query. Say for example, we have a user that wants to create the class "dog". The initial query of this user may be an image of a dog, next to a human. And say that the human takes a large surface of the image. Only relying on this positive image, and a fixed metric to compute similarity will result in images of dogs and/or humans. Seeing that the human is larger on the image, there will be more images with only humans than images with only dogs. The selected samples during this only step are mainly negative, and the positive ones may not even cover different types of dogs! Iterative steps are what allows the user to annotate the images with humans as negative, so that this region is no longer used, and new regions are explored to maybe cover new types of dogs. Of course, the one-step retrieval would be the ideal case, but as shown in our experiment the performance of the first step is too low and can be strongly improved with a few iterations.
>
> **Suggested methods are expensive**
> - When incorporating diversity, we first select a number > k of samples based on MP, then choose the k most diverse from these samples. We used 50 samples, and our experiences showcased an increase in the wait times. So the solution is feasible, but it is more expensive than ours, and does not give better results.
> - SIMILAR and Core-Set aim to cover the feature space of all the unlabeled data, which is known, in order to visit all classes. We aim to only cover the feature subspace of our class, as we only want to retrieve our class-of-interest. We have no idea about this feature subspace beforehand.
> - I agree, SIMILAR can be used on standard features. But the the complexity of the greedy algorithm remains very high.
>
> **No priors about the class**
> - Yes, the query is the prior. We agree with the reviewer that applying SIMILAR on our problem results in solving $argmax_A I_f(A;P/N)$ where P is the positive example of the query, and N the set of negative samples. However, seeing that the same positive image may contain, next to the class-of-interest, another class, covering the desired class from this only image, is basically trying to predict which class of the two present classes is desired by the user. Also, we have developed SIMILAR for our NCR problem, and trying solving it using in a greedy manner as suggested in the SIMILAR article, using the FLCMI formula. However the $O(Bn²)$ complexity, with n of the order of only a hundred thousand images (not even millions which is the desired use cas), makes the computation extremely slow (25 iterations with PF-MA take less than a minute, whereas selecting 1 sample with SIMILAR takes hours, using the same computational resources).

---

> ### Author Response · Authors · 2025-02-05
> **Response to Response to Rebuttal**
>
> (continued)
>
> **Weak evaluation metric**
> - While we rely on clustering, we do not seek a perfect clustering of our class. This is in fact why we average on a number of clusterings. We only want to topologically divide the class into subspaces and check whether they are all covered. We agree that chosing the number of clusters is not a prior, but we have tested on different values, and concluded that the behavior of the coverage metric is the same. We chose k=32 in order to ensure a large proportion of classes (>70%) is clustered.
> - Normalizing the pairwise distance metric computed on the positive selected samples by the metric computed on the samples of the class may be a good idea. However, it still gives an advantage to samples that are further away, even if closer samples indeed cover different subconcepts of the class. Say a class is composed of 3 different subconcepts A, B and C, two of which (A and B) are close in the feature space. If we select one sample from A, one from B and two from C, we will have a better coverage according the suggested metric than if we select one from A, two from B and one from C. All in all, pairwise distances will always suffer from the non uniformity of the classes.
>
> **PF-MA encourages diversity**
> - The coverage is not only computed on the positive selected samples during a certain iteration, but on the union of positive selected samples from iteration 1. The diversity is in fact a result of returning ambiguous samples, i.e. exploring a certain near boundary region during iteration i, then moving on to a completely different region during iteration i+1, as the previous region is now known. These ambiguous samples being positive does not change that, only enhances the retrieval, and allows not only the exploration of the space, but also the exploitation of the classifier scores. Yes, there is nothing preventing to return an image and its near duplicate during i, based on their uncertainty scores, but this guarantees that 1) this region will not be visited during the next iteration as it is now a known region, 2) these images can never be near duplicates of the positives we have already trained on.
> - As mentioned before, SIMILAR are Core-Set try to span the whole set of unlabeled data, which we can't do with our unknown class, and MMR is basically MP-D, which does not outperform PF-MA. With our method, we do not need to add explicit diversity computation, but still manage to achieve good diversity performances.

---

### Decision · Action_Editor_kPvU · 2025-03-05

**Recommendation:** Reject

**Comment:**

This submission proposes a new active learning method for the problem of “Novel Class Retrieval” (NCR). The method is based on an iterative "human in the loop" procedure. The method is based on iteratively obtaining examples from the user and classifying them using a standard method such as SVM.

This paper does not satisfy the TMLR acceptance criteria of Claims and Evidence and Audience. Please see above the detailed comments for each of these criteria.

**Audience:**

The proposed method seems simplistic, and may not be sufficiently of interest to relevant audience.

**Claims And Evidence:**

The NCR problem and its goal are not clearly defined. The differences from Active Learning are not well explained.

Some of the claims are not supported in the paper. The paper claims that iterative selection is crucial to the success of this but there are no studies showing that existing retrieval strategies (that select all samples in one go) cannot perform on par with the proposed method. Since latency is a concern in this scenario, iterative selection should be avoided unless it is absolutely critical.

Comparison with strong baseline models is missing.